# TOWARDS THE UNIVERSAL LEARNING PRINCIPLE FOR GRAPH NEURAL NETWORKS

## ABSTRACT

Graph neural networks (GNNs) are currently highly regarded in graph representation learning tasks due to their significant performance. Although various propagation mechanisms and graph filters were proposed, few works have considered the convergence and stability of graph filters under infinite-depth scenarios. To address this problem, we elucidate the criterion for the graph filter formed by power series and further establish a scalable regularized learning principle, which can guide us on how to design infinite deep GNN. Following the framework, we develop Adaptive Power GNN (APGNN), a deep GNN that employs exponentially decaying weights to aggregate graph information of different orders so as to mine the deeper neighbor information. Different from existing GNNs, APGNN can be seamlessly extended to an infinite-depth network. Moreover, we analyze the generalization of the proposed learning framework via uniform convergence and present its upper bound in theory. Experimental results show that APGNN obtains superior performance against the state-of-the-art GNNs.

## 1 INTRODUCTION

Recently, Graph Neural Networks (GNNs) have shown commendable performance on numerous graph representation learning tasks. In addition, GNNs have been introduced in a variety of application tasks, such as recommendation systems (Han et al., 2022; Zorzi et al., 2022; Giuliari et al., 2022), computer vision (Deng et al., 2022; Pang et al., 2022; He et al., 2020), and traffic forecasting (Guo et al., 2019; 2021). The fundamental part of GNN is the design of the propagation mechanism or the graph filter (Xu et al., 2019; Feng et al., 2022; Wang and Zhang, 2022a; He et al., 2021; Sandryhaila and Moura, 2013a). GNNs can be categorized into two groups based on the approach of formulation. Spatial-based GNNs design propagation mechanisms through direct aggregation of spatial features. For instance, Graph Convolutional Networks (GCNs) (Kipf and Welling, 2017) aggregate one-hop information on the graph, Graph Attention Networks (GATs) (Veličković et al., 2018) learns node relationships using an attention mechanism and GraphSAGE (Hamilton et al., 2017) employs various pooling operations as aggregation functions.

Spectral-based GNN designs graph filters in graph Fourier domain to discover a proper transformation of the graph spectrum. For example, ChebNet (Defferrard et al., 2016) constructs the localized graph filter with Chebyshev polynomial, PPNP (Klicpera et al., 2019) employs Personalized PageRank to design graph filter and GNN-LF/HF (Zhu et al., 2021) develops the graph filter through a graph optimization framework. In recent years, there has been a growing interest in the notion of learnable polynomial graph filters due to their ability to learn proper graph filters to address both heterophilic and homophilic graphs. Therefore, numerous methods have been proposed utilizing various polynomial bases such as monomial basis (Chien et al., 2021), Bernstein basis (He et al., 2021) and Jacobi basis (Wang and Zhang, 2022b).

Although progress has been made, there are still some limitations on the depth of most GNNs. That is, as the depth approaches infinity, the convergence of graph filters can not be guaranteed. Moreover, no general rule has been explored in previous research to uncover the construction principle of an infinite deep GNN. Motivated by the convergence of power series, we explore how to design a graph filter for constructing an infinite deep GNN. A universal learning principle is then proposed to summarize the rule for designing a graph filter. With this principle, we propose Adaptive Power Graph Neural Network (APGNN), which adaptively learns the task-specific graph filter for node

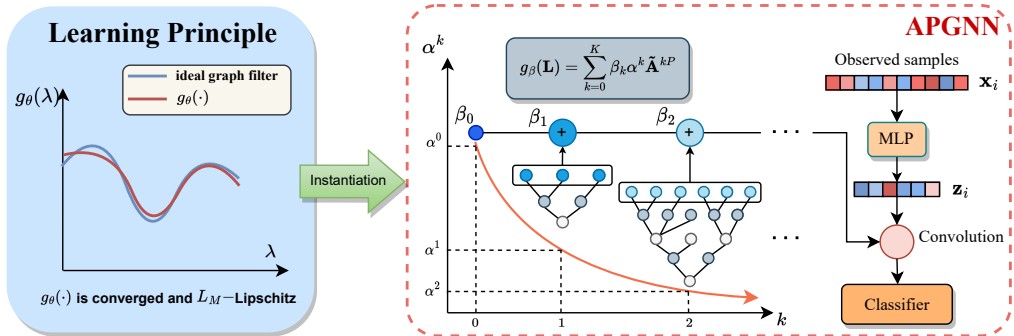

Figure 1: An illustration of the proposed APGNN that adheres to the learning principle. The model incorporates the decay rate $\alpha$ to suppress the information from high-order neighbors while adaptively learning bounded coefficients $\beta$. Furthermore, it aggregates information with $P$-hop to perceive the higher-order neighborhoods. This design enables the seamless extension of APGNN to an infinite deep network.

representation learning. The main idea of APGNN is depicted in Figure 1. The parameterized graph filter is designed with regularization of the exponential decay rate, guaranteeing convergence of the graph filter. A $P$-hop filter is applied to aggregate high-order neighbor information with fewer parameters. Furthermore, the generalization bound of the proposed learning principle is presented with the setting of the continuous graph, which guarantees the generalization ability theoretically.

Our key contributions are summarized as follows: 1) We propose a learning principle with Lipschitz constraints and convergence guarantee on graph filters, providing theoretical guidance for constructing deeper GNN. Following this principle, APGNN is proposed with a graph filter employing exponentially decaying weights along with the order. 2) To investigate the capacity of the proposed framework, we present the generalization analysis and give the upper bound of generalization in theory. 3) Experimental results demonstrate the superiority of the APGNN against the related SOTA methods, showing the effectiveness of the proposed framework.

## 2 PRELIMINARIES

**Notations.** Suppose we have an undirected graph $\mathcal{G} = (\mathcal{V}, \mathcal{E}, \mathbf{A})$ with node set $\mathcal{V}$ and $|\mathcal{V}| = n$. $\mathbf{A} \in \mathbb{R}^{n \times n}$ denotes the adjacency matrix indicating the edges in $\mathcal{E}$. Assuming that the self-loops are contained in the graph, i.e., $a_{ii} = 1$. Let $\mathbf{X} = [\mathbf{x}_1, \mathbf{x}_2, \cdots, \mathbf{x}_n]^\top \in \mathbb{R}^{n \times d}$ be the graph signals (or features) of the nodes. We use notation $[n] \triangleq \{1, 2, \cdots, n\}$ for $n \in \mathbb{N}_+$. Assume that the label of $\mathbf{x}_i$ is $y_i \in \mathcal{Y}$ for all $i \in [n_l]$, where $n_l \leq n$ is the number of labeled samples.

**Graph Neural Networks.** We introduce some essential concepts in GNNs. Let $d_i = \sum_{j=1}^{n} A_{ij}$ be the degree of $i$-th node, so the degree matrix of $\mathbf{A}$ can be defined as $\mathbf{D} = \mathrm{diag}(d_1, d_2, \cdots, d_n)$. The symmetrically normalized Laplacian is $\mathbf{L} = \mathbf{I} - \tilde{\mathbf{A}}$, where $\tilde{\mathbf{A}} \triangleq \mathbf{D}^{-1/2}\mathbf{A}\mathbf{D}^{-1/2}$ is normalized adjacency matrix. Consider the eigen-decomposition $\mathbf{L} = \mathbf{U}\mathbf{\Lambda}\mathbf{U}^\top$, where $\mathbf{\Lambda} = \mathrm{diag}(\lambda_1, \cdots, \lambda_n)$ is the diagonal matrix of eigenvalues, and $\mathbf{U} = [\mathbf{u}_1, \cdots, \mathbf{u}_n]$ represents the eigenvectors associated with the eigenvalues. Note that $\tilde{\mathbf{A}}$ shares the same eigenvectors with $\mathbf{L}$.

Spectral convolution on graphs is defined as the following transformation (Kipf and Welling, 2017; Sandryhaila and Moura, 2013b):

$$g * \mathbf{X} = \mathbf{U}g(\mathbf{\Lambda})\mathbf{U}^\top\mathbf{X}, \tag{1}$$

where $g(\cdot) : [0, 2] \mapsto \mathbb{R}$ is called filter function and $g(\mathbf{\Lambda}) = \mathrm{diag}(g(\lambda_1), \cdots, g(\lambda_n))$. The common approach in GNNs is to apply polynomial functions as the filters (Kipf and Welling, 2017; He et al., 2021; Defferrard et al., 2016), which leads to $\mathbf{U}g(\mathbf{\Lambda})\mathbf{U}^\top = g(\mathbf{L})$. Therefore, spectral convolution is usually written as $g * \mathbf{X} = g(\mathbf{L})\mathbf{X}$. The graph representation paradigm in GNN is generally expressed

as follows:

$$\mathbf{Z} = g(\mathbf{L})f(\mathbf{X}), \quad g(\mathbf{L}) = \sum_{k=0}^{K} \theta_k (\mathbf{I} - \mathbf{L})^k, \tag{2}$$

where $\mathbf{Z} \in \mathbb{R}^{n \times c}$ denotes the node representation, and $f(\cdot)$ represents a feature extractor such as multi-layer perceptions (MLPs).

## 3 RELATED WORKS

### 3.1 LEARNABLE POLYNOMIAL GRAPH FILTER

The establishment of a spectral-based GNN entails the construction of either a fixed graph filter or a learnable graph filter. In the context of fixed graph filters, PPNP employs Personalised PageRank to formulate the graph filter. GNN-LF/HF (Zhu et al., 2021) constructs the graph filter through a graph optimization framework. In terms of the latter construction, GPR-GNN (Chien et al., 2021) applies Generalized PageRank to define the graph filter, which can be seen as a learnable polynomial graph filter. Several studies exhibit a similar idea, employing distinct polynomial bases and imposing different constraints on the coefficients (Defferrard et al., 2016; He et al., 2021; Wang and Zhang, 2022b). These works perform well on both heterophilic and homophilic datasets and also demonstrate the capability to learn appropriate graph filters. However, they did not specify the general principle for constructing graph filters approximating infinite depth. Therefore, it is necessary for us to clarify the universal principle to construct GNNs particularly when the depth approaches infinite.

### 3.2 GENERALIZATION ANALYSIS ON GNNS

Generalization analysis on GNNs has been extensively studied recently. (Esser et al., 2021) and (Tang and Liu, 2023) present the generalization with transductive Rademacher complexity on node classification tasks. Their generalization error is only measured over the testing set. In contrast, (Cong et al., 2021) analyzes the transductive uniform stability of GNN (this is also related to (Verma and Zhang, 2019)). Considering the stochastic hypothesis, (Ma et al., 2021) uses PAC-Bayesian theorem to analyze the subgroup generalization bound of GNN. Moreover, (Li et al., 2022a) and (Zhang et al., 2023) investigate the generalization guarantee of GNN via topology properties in the graph. Different from the previous works, we explore the generalization from the perspective of the continuous graph, which provides the generalization guarantee over the whole sample space.

## 4 LEARNING PRINCIPLE FOR GNNS

### 4.1 THE PRINCIPLE OF DEVISING GRAPH FILTERS

Current studies suggest a significant relationship between the performance of GNN and its graph filter Klicpera et al. (2019); Liu et al. (2020). In general, the general graph filters are characterized by polynomials associated with the adjacency matrix $\tilde{\mathbf{A}}$ (or Laplacian matrix $\mathbf{L}$), i.e., $g(\mathbf{L}) = \sum_{k=0}^{K} \theta_k \tilde{\mathbf{A}}^k$. However, the existing methods still encounter the issue that the depth of GNN is limited. The reason for this phenomenon is that these GNNs are inconsistent with their "infinite-depth" version. That is, the corresponding graph filter can not even converge as the depth $K \to \infty$. Furthermore, it is also uncertain that the learned graph filter is stable. Consequently, the depth of the models is restricted. To address this issue, it is necessary to study the properties of GNNs with infinite depth. Therefore, we explore the graph filter formulated as power series:

$$g(\tilde{\mathbf{A}}) = \sum_{k=0}^{\infty} \theta_k \tilde{\mathbf{A}}^k = \sum_{k=0}^{\infty} \theta_k (\mathbf{I} - \mathbf{L})^k. \tag{3}$$

First of all, a well-defined graph filter represented as equation (3) must be convergent. Therefore, it becomes essential to investigate what kind of properties the coefficients $\theta_k$ should have. The following lemma provides sufficient and necessary conditions for the coefficients of the graph filter.

**Lemma 1.** *Let $\{a_k\}$ and $\{\gamma^k\}$ be the real number sequences, where $\gamma \in (-1, 1]$ and $k \in \mathbb{N}$. Then $\sum_k^{\infty} a_k \gamma^k$ converges uniformly and absolutely if and only if the series $\sum_k^{\infty} a_k$ converges absolutely.*

As a direct result, the coefficients of the graph filter (i.e., $\theta_k$) should satisfy the following theorem.

**Theorem 1.** *Let $\tilde{\mathbf{A}} = \mathbf{D}^{-1/2}\mathbf{A}\mathbf{D}^{-1/2}$ be the normalized adjacency matrix of a graph $\mathcal{G}$ with spectral radius $\|\tilde{\mathbf{A}}\|_2 \leq 1$. The matrix series $\sum_{k=0}^{\infty} \theta_k \tilde{\mathbf{A}}^k$ converges uniformly and absolutely if and only if the series $\sum_{k=0}^{\infty} \theta_k$ converges absolutely.*

The proofs of Lemma 1 and Theorem 1 are shown in Appendix. Theorem 1 offers a sufficient and necessary condition for the convergence of graph filters formed by power series. Specifically, the condition requires the existence of a finite real number $M \geq 0$, such that

$$\|\boldsymbol{\theta}\|_1 \triangleq \sum_{k=0}^{\infty} |\theta_k| \leq M. \tag{4}$$

Therefore, an arbitrary graph filter formed by power series should satisfy the above convergence condition, which gives the first requirement when designing GNN. Apart from convergence, we expect the graph filter to possess good analytic properties such as stability. To this end, Lipschitz continuity should be considered as the second requirement of the graph filter. Let $g(\cdot)$ be an $L$-Lipschitz continuous function, meaning that

$$|g(\lambda) - g(\lambda')| \leq L|\lambda - \lambda'|, \quad \forall \lambda, \lambda' \in [0, 2). \tag{5}$$

This property indicates the stability or robustness of the model (Gama et al., 2020; Pauli et al., 2021). If the graph is contaminated and its eigenvalues are perturbed by at most $\epsilon$, Lipschitz continuity ensures that the perturbation of the graph-filtered result is at most $L\epsilon$. For instance, considering $g(\lambda) = \sum_{k=0}^{\infty}(1-\lambda)^k/k^2$, which is convergent, yet the Lipschitz condition does not satisfy for $\lambda$ closed to zero. Therefore, this graph filter might be sensitive to the input graph. Subsequently, we conclude the following criterion for designing GNN.

$$\mathbf{Z} = g_{\boldsymbol{\theta}}(\mathbf{L})f(\mathbf{X}), \text{ with } \|\boldsymbol{\theta}\|_1 \leq M, \ g_{\boldsymbol{\theta}}(\cdot) \text{ is a Lipschitz function.} \tag{6}$$

To enhance the scalability of the model, we define $\boldsymbol{\theta}$ as a learnable parameter (though its dimension is infinite). In this way, (6) gives a regularized learning framework for GNN. Therefore, for any sufficient large $K \in \mathbb{N}_+$ and the $K$-order polynomial graph filter $g_{\boldsymbol{\theta}}^K(\lambda) = \sum_{k=0}^{K} \theta_k(1-\lambda)^k$, the condition (6) should be satisfied for some $K$-inpendent constant $M > 0$, which keeps the consistency with its infinitely deep version $g_{\boldsymbol{\theta}}^{\infty}(\lambda) = \sum_{k=0}^{\infty} \theta_k(1-\lambda)^k$. We will present the applications of this criterion in the following section, and further analyze its generalization in section 5.

## 4.2 CONNCECTIONS WITH EXISTING GNNS UNDER THE LEARNING PRINCIPLE

In this subsection, we investigate the relationship between our learning principle and several well-known Graph Neural Networks (GNNs), focusing on the design of graph filters. Our findings indicate that these GNNs are all special cases of our learning principle.

**PPNP** (Klicpera et al., 2019). PPNP uses Personalized PageRank as the graph filter, which balances the local information preservation and the high-order neighbor information. The model of PPNP is $\mathbf{Z} = \alpha(\mathbf{I} - (1-\alpha)\tilde{\mathbf{A}})^{-1}\mathbf{H} = (\mathbf{I} + \beta\mathbf{L})^{-1}\mathbf{H}$, where $\mathbf{H} = f(\mathbf{X})$ is a two-layer MLPs and $\beta = 1/\alpha - 1$. Hence, the graph filter of PPNP is $g_{\text{PPNP}}(\mathbf{L}) = (\mathbf{I} + \beta\mathbf{L})^{-1}$. Considering its Taylor series, we have

$$g_{\text{PPNP}}(\mathbf{L}) = (\mathbf{I} + \beta\mathbf{L})^{-1} = \frac{1}{1+\beta}\sum_{k=0}^{\infty}\left(\frac{\beta}{1+\beta}\right)^k \tilde{\mathbf{A}}^k = \sum_{k=0}^{\infty} \theta_k \tilde{\mathbf{A}}, \tag{7}$$

where $\theta_k = \beta^k/(1+\beta)^{k+1}$. It is straightforward to validate that $\sum_{k=0}^{\infty} \theta_k = 1$, and thus the convergence requirement (4) holds. Moreover, the Lipschitz condition is easily verified. Thus PPNP satisfies the criterion of (6). However, the performance of PPNP is heavily dependent on the hyperparameter $\beta$, which must be carefully tuned to achieve the optimal performance.

**DAGNN** (Liu et al., 2020). DAGNN adaptively adjusts the weight of information aggregation from different neighbors to solve the over-smoothing problem. It designs a parameterized graph filter formulated as a $K$-order polynomial:

$$g_{\text{DAGNN}}(\mathbf{L}) = \sum_{k=0}^{K} \theta_k \tilde{\mathbf{A}}^k, \quad \text{s.t. } 0 \leq \theta_k \leq 1, \tag{8}$$

where $\theta_k$ is the learnable parameter with bounded constraint. Due to this adaptive learning strategy, DAGNN is able to learn a graph filter more suitable for node classification. The empirical studies suggest DAGNN works well with a proper $K$. However, as $K \to \infty$, the constraint $0 \le \theta_k \le 1$ cannot guarantee the convergence of the graph filter. It indicates that DAGNN is "inconsistent" with its infinitely deep version. Therefore, it can not be naturally extended to infinity deep GNN.

**GPR-GNN** (Chien et al., 2021). GPR-GNN introduced truncated Generalized PageRank architecture, which is equivalent to a $K$-order polynomial graph filter, for topological information extraction. That is:

$$g_{\text{GPR}}(\mathbf{L}) = \sum_{k=0}^{K} \theta_k \tilde{\mathbf{A}}^k, \quad \text{s.t.} \sum_{k=0}^{K} \theta_k = 1. \tag{9}$$

Under this constraint, the learned parameter $\theta_k$ is permitted to have negative values, which allows for the preservation of relevant high frequencies and enables good performance on heterophilic graph datasets. When $K \to \infty$, $g_{GPR}$ is converged and the Lipschitz continuity can be verified. Therefore, GPR-GNN adheres to 6 and can be extended to infinite depth.

### 4.3 INSTANTIATION: ADAPTIVE POWER GRAPH NEURAL NETWORK

We begin to introduce a novel GNN following the principle proposed in section 4.1, called Adaptive Power GNN (APGNN). We first consider the following graph filter parameterized by $\boldsymbol{\beta}$ with the form:

$$g_{\boldsymbol{\beta}}^{\infty}(\lambda) = \sum_{k=0}^{\infty} \beta_k \alpha^k (1-\lambda)^k, \quad \text{where } |\beta_k| \le 1, \ 0 < \alpha < 1, \tag{10}$$

where the coefficient of the power series $\theta_k = \beta_k \alpha^k$, with hyper-parameter $\alpha \in (0, 1)$ ensuring the convergence. Immediately, we check the condition of Lemma 1.

$$\|\boldsymbol{\theta}\|_1 = \sum_{k=0}^{\infty} |\beta_k \alpha^k| \le \sum_{k=0}^{\infty} \alpha^k \le \frac{1}{1-\alpha}. \tag{11}$$

Hence, the power series converges on $[0, 2]$ absolutely and uniformly. Similarly, the associated matrix series $g_{\boldsymbol{\beta}}^{\infty}(\mathbf{L}) = \sum_{k=0}^{\infty} \beta_k \alpha^k \tilde{\mathbf{A}}^k$ also converges uniformly and absolutely by Theorem 1. Moreover, $g_{\boldsymbol{\beta}}^{\infty}(\cdot)$ is $\alpha(1-\alpha)^{-2}$-Lipschitz. To see this, for any $|\beta_k| \le 1$ and $1 - \lambda \in (-1, 1]$, we have

$$|\nabla g_{\boldsymbol{\beta}}^{\infty}(\lambda)| = \left| \sum_{k=1}^{\infty} k \beta_k \alpha^k (1-\lambda)^{k-1} \right| \le \sum_{k=1}^{\infty} k\alpha^k = \frac{\alpha}{(1-\alpha)^2}, \tag{12}$$

which implies the Lipschitz continuous property. Thus, this graph filter fits the requirement of the proposed criterion. However, the model with this graph filter is unavailable in practice as the number of parameters to be learned is infinite. The $K$-order truncated polynomial is utilized for substitution, i.e., $g_{\boldsymbol{\beta}}^K(\mathbf{L}) = \sum_{k=0}^{K} \beta_k \alpha^k \tilde{\mathbf{A}}^k$. We evaluate the approximation via the upper bound of $K$-order truncation error:

$$|g_{\boldsymbol{\beta}}^{\infty}(\lambda) - g_{\boldsymbol{\beta}}^K(\lambda)| \le \sum_{k=K+1}^{\infty} |\beta_k \alpha^k (1-\lambda)^k| \le \sum_{k=K+1}^{\infty} \alpha^k = \frac{\alpha^{K+1}}{1-\alpha}, \tag{13}$$

which uniformly holds for $\forall \lambda \in [0, 2]$. Likewise, the approximation error of matrix series is given by

$$\left\| g_{\boldsymbol{\beta}}^{\infty}(\mathbf{L}) - g_{\boldsymbol{\beta}}^K(\mathbf{L}) \right\|_2 = \left\| \mathbf{U} \left( g_{\boldsymbol{\beta}}^{\infty}(\mathbf{\Lambda}) - g_{\boldsymbol{\beta}}^K(\mathbf{\Lambda}) \right) \mathbf{U}^{\top} \right\|_2 = \sup_{i \in [n]} |g_{\boldsymbol{\beta}}^{\infty}(\lambda_i) - g_{\boldsymbol{\beta}}^K(\lambda_i)| \le \frac{\alpha^{K+1}}{1-\alpha}, \tag{14}$$

where $\lambda_i$ denotes the $i$-th eigenvalue of $\mathbf{L}$. This upper bound is independent of the given graph, which can be controlled via tuning $\alpha$ and $K$. The higher $K$ and smaller $\alpha$ yield a better approximation to the exact graph filter $g_{\boldsymbol{\beta}}^{\infty}(\cdot)$. Nevertheless, the small $\alpha$ tends to limit the capability of the graph filter. Extremely, $\alpha \to 0$ gives a trivial function $g_{\boldsymbol{\beta}}^K(\lambda) = \beta_0$. This suggests that $\alpha$ should be elaborately tuned to improve the performance.

Though the aforementioned graph filter is primarily motivated via spectral analysis, we can still present the spatial view explanation for its design. Existing GNNs aggregate the neighbor information

of different hops with certain weights, which could be either manually assigned or learned adaptively. Typically, methods like GPR-GNN (Chien et al., 2021) and DAGNN (Liu et al., 2020), which can learn the aggregation weight, tend to treat the neighbor's information of different hops equally. That is, the $k$-layer's weight are assigned with $\theta_k = \mathcal{O}(1)$ for each $k \in [K]$. However, it is shown in the previous research that the propagation with the very high-order neighbor potentially leads to the over-smoothing issue (Wu et al., 2019; Rong et al., 2020). The current methods magnify this flaw of the high-order graph since they cannot distinguish the significance of the information of different hops. This motivates the design of the decay rate in APGNN, i.e., we employ weights with exponential decaying rate by assigning $\theta_k = \mathcal{O}(\alpha^k)$ for some $0 < \alpha < 1$. This approach emphasizes the contribution of lower-order neighbors and restricts the over-weighting of the information from high-order neighbors due to $\theta_k \to 0$ with $k \to \infty$. Therefore, it provides more effective aggregation and thus enhances the model's performance.

Furthermore, we discuss a general formulation of graph filter, i.e, $g_{\boldsymbol{\beta}}^{K,P}(\lambda) = \sum_{k=0}^{K} \beta_k \alpha^k (1-\lambda)^{kP}$, called $P$-hop filter. It is obvious that it follows the proposed learning principle. Intuitively, the $P$-hop filter tends to perceive deeper neighbor information (up to $(KP)$-th order graph). From the spectral view, it provides an effective way to reduce the number of parameters $K$. There exists a $\delta > 0$ such that all non-zero eigenvalues of $\mathbf{L}$ satisfies $\lambda_i \in [\delta, 2 - \delta]$. For these eigenvalues we have $|g_{\boldsymbol{\beta}}^{\infty,P}(\lambda_i) - g_{\boldsymbol{\beta}}^{K,P}(\lambda_i)| < (\alpha(1-\delta)^P)^{K+1}/(1-\alpha)$, where $g_{\boldsymbol{\beta}}^{\infty,P}(\cdot) = \lim_{K \to \infty} g_{\boldsymbol{\beta}}^{K,P}(\cdot)$. Thus, we need $K \geq \mathcal{O}(P^{-1} \log_{1-\delta} \varepsilon)$ to reach the approximation precision $\varepsilon$. Compared with the uniform bound (13) that implies $K \geq \mathcal{O}(\log(1/\varepsilon))$, this result suggests that $K$ can be reduced by increasing $P$. Also note that the Lipschitz constant of $g_{\boldsymbol{\beta}}^{\infty,P}(\cdot)$ is $P\alpha/(1-\alpha)^2$. Therefore, $P$ should not be excessively large to ensure the stability of the graph filter. The empirical studies also demonstrate that the $P$-hop filter enhances the performance of APGNN. See section 6.2 for details.

Summarizing the above analysis, we present the following comprehensive architecture of APGNN:

$$\mathbf{Z} = g_{\boldsymbol{\beta}}^{K,P}(\mathbf{L}) \text{MLP}(\mathbf{X}), \ g_{\boldsymbol{\beta}}^{K,P}(\mathbf{L}) = \sum_{k=0}^{K} \beta_k \alpha^k \tilde{\mathbf{A}}^{kP}, \text{where } |\beta_k| \leq 1, \ 0 < \alpha < 1. \quad (15)$$

In short, APGNN incorporates the benefits from the decay rate $\alpha$ that exponentially suppresses the information of extremely high-order neighbors and the $P$-hop filter for enlarging the receptive field. These approaches make it possible to realize a sufficiently deep GNN. The computational complexity analysis is shown in Appendix.

## 5 GENERALIZATION ANALYSIS

The previous research focused on discussing the generalization of GNN within the discrete graph with transductive Rademacher complexity (Cong et al., 2021; Verma and Zhang, 2019). In contrast, we analyze the uniform generalization bound of the proposed GNN learning principle under the continuous graph setup.

We first introduce some notations. Denote $\mathbf{x} \in \mathcal{X}$ as any samples from the input space $\mathcal{X}$ (we generally set $\mathcal{X}$ as a subset of $\mathbb{R}^d$). Let $p(\cdot)$ be a probability measure defined over $\mathcal{X}$. Assume $x_j$ is the $j$-th coordinate of $\mathbf{x} \in \mathcal{X}$ and $\mathbb{E}[x_j^2] \leq c_{\mathcal{X}}^2$ for any $j \in [d]$, where $c_{\mathcal{X}}$ is a constant dependent on data. To describe the graph relation between each pair $(\mathbf{x}, \mathbf{x}')$ over $\mathcal{X} \times \mathcal{X}$, we define a continuous graph function $A(\cdot, \cdot) : \mathcal{X} \times \mathcal{X} \mapsto \mathbb{R}_+$, and its corresponding degree function is

$$d(\mathbf{x}') = \int_{\mathcal{X}} A(\mathbf{x}, \mathbf{x}') \mathrm{d}p(\mathbf{x}'). \quad (16)$$

Different from the setting in (Rosasco et al., 2010; Li et al., 2022b), we assume $0 \leq A(\mathbf{x}, \mathbf{x}')$, and $0 \leq d(\mathbf{x})$ for any $\mathbf{x}, \mathbf{x}' \in \mathcal{X}$. Therefore, we can define the symmetric normalized graph:

$$\tilde{A}(\mathbf{x}, \mathbf{x}') = \frac{A(\mathbf{x}, \mathbf{x}')}{\sqrt{d(\mathbf{x})d(\mathbf{x}')}}. \quad (17)$$

Then the corresponding normalized Laplacian is $L = I - \tilde{A}$, where $I$ indicates the identity operator over $\mathcal{X}$. For a graph filter function $g_{\boldsymbol{\theta}}(\lambda) = \sum_{k=0}^{K} \theta_k (1-\lambda)^k$, graph convolution of the continuous

graph is defined as the following integral operator:

$$g_{\boldsymbol{\theta}} L f = \sum_{k=0}^{K} \theta_k \tilde{A}^k f, \quad \tilde{A} f = \int_{\mathcal{X}} \tilde{A}(\cdot, \mathbf{x}) f(\mathbf{x}) \mathrm{d} p(\mathbf{x}),$$  (18)

where $\tilde{A}^k = \tilde{A}^{k-1} \circ \tilde{A}$ denotes $k$-order composition of integral operator with $\tilde{A}^0 = I$. Note we have $\sum_{k=0}^{K} \theta_k \|\tilde{A}\| \leq \|\boldsymbol{\theta}\|_1 \leq M$ for any $K \in \mathbb{N}$, indicating $\sum_{k=0}^{\infty} \theta_k \tilde{A}$ is absolutely summable. This guarantees the existence of graph filter on the continuous graph when $K \to \infty$. For convenience in understanding, we provide the analysis on a simplified GNN, where we consider a semi-supervised learning task with two classes, i.e., $y_i \in \mathcal{Y} \triangleq \{-1, 1\}$, and utilize linear feature extractor $f(\mathbf{X}) = \mathbf{w}^\top \mathbf{X}$. Note that we can still extend our result for $f(\mathbf{X}) = \mathrm{MLP}(\mathbf{X})$ and multi-class cases using the techniques proposed in (Bartlett et al., 2017). With the above setting, the hypothesis set over $\mathcal{X}$ is described as

$$\mathcal{H}_{\mathcal{X}} = \{h : h(\mathbf{x}) = g_{\boldsymbol{\theta}} L f(\mathbf{x}), \ f(\mathbf{x}) = \langle \mathbf{w}, \mathbf{x} \rangle, \ \|\mathbf{w}\|_2 \leq B, \ \|\boldsymbol{\theta}\|_1 \leq M\}.$$  (19)

However, the integral in each hypothesis $h \in \mathcal{H}_{\mathcal{X}}$ is intractable since the underlying graph function and the data distribution are unknown. Therefore, we should use the "empirical version" of the hypothesis to estimate $h \in \mathcal{H}_{\mathcal{X}}$. For this reason, we introduce the hypothesis set defined over the observed samples $S$ and graph $\mathcal{G}$:

$$\mathcal{H}_S = \left\{ h : h(\mathbf{x}_i) = \sum_{j=1}^{n} g_{\boldsymbol{\theta}}(\mathbf{L})_{ij} \mathbf{x}_j^\top \mathbf{w}, \quad \|\mathbf{w}\|_2 \leq B, \ \|\boldsymbol{\theta}\|_1 \leq M \right\}.$$  (20)

Define the generalization error and the empirical error Mohri et al. (2018) as follows

$$R(h) = \mathbb{E}_{(\mathbf{x}, y)}[\mathbf{1}_{yh(\mathbf{x}) \leq 0}], \quad \hat{R}(h) = \frac{1}{n_l} \sum_{i=1}^{n_l} \min(1, \max(0, 1 - y_i h(\mathbf{x}_i))).$$  (21)

We have the following theorem on the generalization of the proposed learning paradigm.

**Theorem 2.** *Suppose $g_{\boldsymbol{\theta}}(\cdot)$ is $L_M$-Lipschitz. Let $h_{\mathbf{w}, \boldsymbol{\theta}} \in \mathcal{H}_{\mathcal{X}}$ and $h_{\mathbf{w}, \boldsymbol{\theta}} \in \mathcal{H}_S$ share the same parameter $(\mathbf{w}, \boldsymbol{\theta})$. Then there exists a constant $C > 0$ related to the graph function, with the probability at least $1 - \delta$, the following inequality holds.*

$$R(h_{\mathbf{w}, \boldsymbol{\theta}}) \lesssim \hat{R}(\hat{h}_{\mathbf{w}, \boldsymbol{\theta}}) + 2BMc_{\mathcal{X}} \sqrt{\frac{2d \log(2K + 2)}{n_l}} + BCL_M dc_{\mathcal{X}} \sqrt{\frac{\log(2/\delta)}{n}}.$$  (22)

The proof is given by excess risk decomposition, shown in Appendix. The notation "$\lesssim$" denotes "less than or approximately equal to the right-hand side" and guarantees an approximation error of at most $\mathcal{O}(\sqrt{\frac{\log(1/\tau)}{n_l}})$ with a probability of at least $1 - \mathcal{O}(\tau)$. We remind readers the important difference between $R(h_{\mathbf{w}, \boldsymbol{\theta}})$ and $\hat{R}(\hat{h}_{\mathbf{w}, \boldsymbol{\theta}})$. The former term measures the population error over the whole input space with the **continuous** graph filter $g_{\boldsymbol{\theta}} L$. In contrast, $\hat{R}(\hat{h}_{\mathbf{w}, \boldsymbol{\theta}})$ is the empirical risk (i.e., training risk) on the sample set $S$ with the **discrete** graph filter $g_{\boldsymbol{\theta}}(\mathbf{L})$. $h_{\mathbf{w}, \boldsymbol{\theta}}$ shares the same learning parameter with $\hat{h}_{\mathbf{w}, \boldsymbol{\theta}}$. Therefore, the minimization of the right-hand-side of (22) w.r.t $(\mathbf{w}, \boldsymbol{\theta})$ reduces the upper bound of the population error.

We observe the first term of generalization bound is of order $\mathcal{O}((dn_l^{-1} \log K)^{1/2})$, which outlines the model's complexity. Although it becomes infinity when $K \to \infty$, the growth of this term is extremely slow as $K$ increases. In practice, we generally set $K < n$ since the neighbor information beyond $n$-hops is redundant, restricting the complexity away from infinity. Therefore, the generalization of the model is rigorously guaranteed for sufficiently large $K$, which allows us to construct significantly deep GNN in the proposed principle. In the following proposition, we unveil the generalization of APGNN as a direct application of Theorem 2.

**Proposition 1.** *Let $\boldsymbol{\beta} \in \mathbb{R}^K$ and $g_{\boldsymbol{\beta}}^K(\lambda) = \sum_{k=0}^{K} \beta_k \alpha^k (1 - \lambda)^k$ where $0 < \alpha < 1$ and $\|\boldsymbol{\beta}\|_\infty \leq 1$. with the probability at least $1 - \delta$, the following inequality holds.*

$$R(h_{\mathbf{w}, \boldsymbol{\beta}}) \lesssim \hat{R}(\hat{h}_{\mathbf{w}, \boldsymbol{\beta}}) + \frac{2Bc_{\mathcal{X}}(1 - \alpha^K)}{1 - \alpha} \sqrt{\frac{2d \log(2K + 2)}{n_l}} + \frac{BCdc_{\mathcal{X}}\alpha}{(1 - \alpha)^2} \sqrt{\frac{\log(2/\delta)}{n}}.$$  (23)

Table 1: The average accuracy (%) and standard deviation (%) on eight benchmark datasets. The highest accuracy in each column is shown in bold, while the second-best result is underlined.

| Model | Dataset | | | | | | | |
|---|---|---|---|---|---|---|---|---|
| | Cora | Citeseer | Pubmed | Wiki-CS | MS-Academic | Cornell | Wisconsin | Texas |
| MLP | 57.79±0.11 | 61.20±0.08 | 73.23±0.05 | 65.66±0.20 | 87.79±0.42 | 83.54±3.83 | 84.54±2.34 | 86.22±3.64 |
| ChebNet | 79.92±0.18 | 70.90±0.37 | 76.98±0.16 | 63.24±1.43 | 90.76±0.73 | 79.42±3.83 | 86.08±2.67 | 84.32±3.13 |
| GCN | 82.03±0.27 | 71.05±0.33 | 79.26±0.18 | 72.05±0.45 | 92.07±0.13 | 59.34±2.85 | 63.89±3.02 | 70.08±3.53 |
| SGC | 81.89±0.26 | 72.18±0.24 | 78.58±0.15 | 72.76±0.35 | 89.01±0.40 | 61.32±2.32 | 62.54±2.56 | 72.56±3.15 |
| GAT | 82.82±0.36 | 71.96±0.39 | 79.15±0.34 | 74.36±0.58 | 91.86±0.27 | 76.47±2.35 | 60.73±1.91 | 76.47±2.16 |
| PPNP | 83.73±0.31 | 71.74±0.44 | 80.28±0.22 | 74.69±0.53 | 92.58±0.06 | 80.85±2.31 | 73.29±1.18 | 76.85±2.12 |
| APPNP | 83.73±0.21 | 71.70±0.21 | 80.07±0.21 | 74.91±0.61 | 92.81±0.12 | 80.27±2.50 | 72.79±1.91 | 76.65±2.31 |
| GNN-LF(iter) | 83.83±0.36 | 71.44±0.42 | 80.31±0.16 | 75.19±0.49 | 92.78±0.22 | 86.92±1.92 | 92.21±2.21 | 87.69±2.69 |
| GNN-HF(iter) | 83.68±0.31 | 71.58±0.36 | 79.99±0.22 | 74.71±0.55 | 92.72±0.31 | 88.85±1.54 | 92.94±1.62 | 89.62±1.35 |
| DAGNN | 82.70±0.17 | 71.90±0.06 | 80.06±0.30 | 75.63±0.48 | 93.24±0.21 | 77.83±2.83 | 86.27±2.95 | 76.81±2.32 |
| GPRGNN | 82.21±0.51 | 69.95±0.94 | 79.59±0.79 | 75.02±0.62 | 92.03±0.22 | 85.49±3.92 | 82.55±6.23 | 81.35±5.32 |
| BernNet | 80.99±1.27 | 70.01±0.57 | 79.05±1.01 | 75.32±0.54 | 92.52±0.47 | 90.39±1.96 | 91.18±1.47 | **91.74**±2.34 |
| APGNN | **84.15**±0.23 | **72.44**±0.56 | **80.74**±0.24 | **76.03**±0.51 | **93.69**±0.20 | **93.27**±2.69 | **94.12**±1.32 | 91.06±2.12 |

*Proof.* This is a direct result with $M = (1 - \alpha^K)/(1 - \alpha)$ and $L_M = \alpha/(1 - \alpha)^2$ in Theorem 2. □

We can also apply Theorem 2 on the existing method to find the upper bound of generalization error. For DAGNN, we have $M = K$ and the Lipschitz constant $L_M = K(K + 1)/2$. For GPR-GNN, $M = 1, L_M = K$. Therefore, the final two terms of DAGNN and GPR-GNN in equation (22) respectively rely on $O(K\sqrt{\log K}), O(K^2)$ and $O(\sqrt{\log K}), O(K)$. Hense, it tends to show weaker generalization in comparison to APGNN as $K$ increases.

# 6 EXPERIMENT

In this section, we conduct node classification experiments on various benchmark datasets to evaluate the performance of APGNN. Specifically, we compare our method with state-of-the-art methods and display the corresponding learned graph filters on different datasets. Moreover, to validate the theoretical analysis, the influence of parameters $K$, $\alpha$, and $P$ is also investigated in experiments.

## 6.1 EXPERIMENT SETUP

**Datasets.** We perform experiments on eight benchmark datasets commonly used in node classification tasks. **1). Cora, Citeseer, Pubmed (Yang et al., 2016; Sen et al., 2008)**: These are three homophilic benchmark datasets. **2). Cornell, Wisconsin, Texas (Pei et al., 2020)**: Three widely used heterophilic benchmark datasets. **3).Wiki-CS (Mernyei and Cangea, 2020)**: A dataset driven from Wikipedia. This dataset defines the computer science articles as nodes, while the hyperlinks are edges. **4). MS Academic (Klicpera et al., 2019)**: A co-authorship Microsoft Academic Graph, where the nodes are the bag-of-words representation of the papers' abstract and edges are co-authorship. The data statistics and their partitions are presented in Appendix.

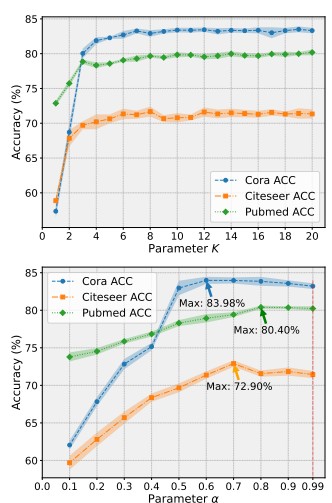

Figure 2: Accuracy with different $K$ in the figure above and different $\alpha$ in the figure below.

**Baselines.** To evaluate the effectiveness of APGNN, we compare it with the following baseline models: 1) MLP (Pal and Mitra, 1992), a traditional method that does not use graphs, 2) GCN (Kipf and Welling, 2017), GAT (Veličković et al., 2018), GraphSAGE (Hamilton et al., 2017) and DAGNN (Liu et al., 2020) spatial methods that aggregate neighborhoods' information, and 3) ChebNet (Defferrard et al., 2016), SGC (Wu et al., 2019), PPNP, APPNP(Klicpera et al., 2019), GNN-LF (iteration form), GNN-HF (iteration form) (Zhu et al., 2021), GPR-GNN (Chien et al., 2021) and BernNet (He et al., 2021) spectral methods analyzing GNNs with graph Fourier transform.

**Settings.** We conducted 10 runs for each method on each dataset, with a hidden dimension of 64. For all compared methods, their parameter settings follow the previous practices (Liu et al., 2020; Zhu et al., 2021). We fix the polynomial order $K$ to 10 in ChebNet, APPNP, GNN-LF, GNN-HF,

DAGNN, APPNP, GPR-GNN and BernNet. The best hyperparameters we choose for APGNN are presented in Appendix. To ensure a fair comparison with the compared methods, we also applied our optimal hyperparameters to them, selecting the maximum value to display.

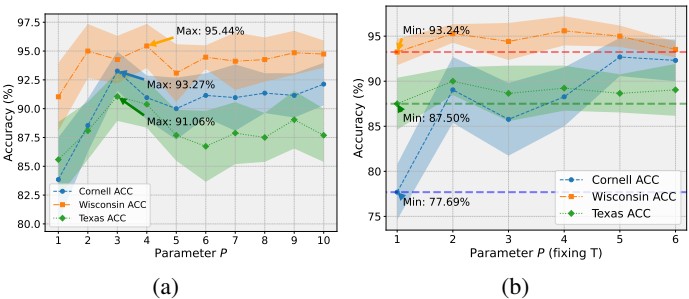

Figure 3: Parameter study on $P$ in two scenarios: (a) fixing $K$ , and (b) fixing $T = KP$.

## 6.2 EXPERIMENTAL ANALYSIS

**Node Classification.** Table 1 reports the average classification accuracy on different datasets, which involves homophilic and heterophilic graphs. We can observe that APGNN achieves the highest accuracy in most cases, demonstrating its superior performance. This reflects that APGNN obtains more effective graph filters than the existing methods and exhibits better generalization ability.

**Polynomial Order $K$.** To gain insight into the role of polynomial order $K$, we conduct the experiment tuning $K$ in $\{1, 2, ..., 20\}$ on Cora, Citeseer, and Pubmed dataset as in the above subfigure within Figure 2. The results show that small $K$ usually results in suboptimal performance, which is because the low-order polynomial cannot sufficiently approximate the underlying ideal filters. It can be observed that the accuracy rate has little promotion for $K > 10$. The reason is the truncation error is adequately small and increasing $K$ might not lead to significant performance enhancement.

**Decay Rate $\alpha$.** The below subfigure within Figure 2 depicts the accuracy curve corresponding to various $\alpha$ values ranging from $0.1$ to $0.99$. We observe that the optimal $\alpha$ generally lies in $[0.6, 0.9]$. The classification performance declines greatly while selecting underestimated $\alpha$ (e.g. $\alpha \leq 0.5$), which tends to cause a trivial filter.

**$P$-hop filter.** We evaluate the influence of changes in $P$ on performance. In Figure 3 (a), we can see that, accuracy increases first and then decreases with the increase of $P$ when fixing $K$. This observation demonstrates that increasing $P$ benefits performance, but a large $P$ will affect stability, consequently leading to accuracy degradation. Moreover, we investigate the accuracy associated with varying parameters $P$ taken from the set $\{1, 2, 3, 4, 5, 6\}$ when fixing the maximum polynomial order $T = KP = 60$. As illustrated in Figure 3 (b), the accuracy is significantly increased when the value of $P$ exceeds 1, particularly in the case of the Cornell dataset. A similar trend is also observed in the datasets such as Cora, Citseer, and Pubmed, as shown in the Appendix. This phenomenon suggests that using the $P$-hop filter enables the model to maintain or even increase its performance while reducing computational costs.

## 7 CONCLUSION

In this work, we propose a universal learning principle for developing convergent and stable GNNs. A practical model named APGNN is proposed to verify the effectiveness of the learning principle. The theoretical analysis of the proposed principle is provided, indicating a stronger generalization ability over the previous works, which is validated by the experimental results. In the future, it is worth exploring diverse graph filters based on the proposed principle. As shown in the generalization analysis, the upper bound of the model complexity relies on $\mathcal{O}(\sqrt{\log K})$. How to design the GNN with complexity free of the hyperparameter $K$ is a meaningful research direction. Lastly, establishing a comprehensive theoretical analysis of the $P$-hop filter is also an important topic.

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

# A  APPENDIX

## A.1  DATA STATISTICS

Table 2: Data statistics for the node classification task.

| Dataset | Nodes | Edges | Features | Class | Train | Val | Test | Homophily level |
|---|---|---|---|---|---|---|---|---|
| cora | 2708 | 5429 | 1433 | 7 | 140 | 500 | 1000 | 0.83 |
| citeseer | 3327 | 4732 | 3703 | 6 | 120 | 500 | 1000 | 0.72 |
| pubmed | 19717 | 44338 | 500 | 3 | 60 | 500 | 1000 | 0.39 |
| wiki-cs | 11701 | 216123 | 300 | 10 | 200 | 500 | 1000 | 0.65 |
| ms academic | 18333 | 81894 | 6805 | 15 | 300 | 500 | 1000 | 0.83 |
| cornell | 183 | 295 | 1703 | 5 | 48% | 32% | 20% | 0.15 |
| wisconsin | 251 | 499 | 1703 | 5 | 48% | 32% | 20% | 0.39 |
| texas | 183 | 309 | 1703 | 5 | 48% | 32% | 20% | 0.1 |

The homophily level is evaluated by the way given in (Chien et al., 2021).

## A.2  LEARNED GRAPH FILTERS

As shown in Figure 4, the prior selection of $P$ affects the orientation of the graph filter. An even $P$ always imposes a graph filter with a symmetric pattern. When $P$ is an even number, $(1 - \lambda)$ with the same absolute value will be evaluated equally, resulting in a symmetric graph filter. In this sense, the low-frequency and high-frequency will be treated equally. In contrast, the odd $P$ tends to derive either a high-pass or low-pass graph filter. Figure 5 and Figure 6 show the detail shape of the learned graph filters.

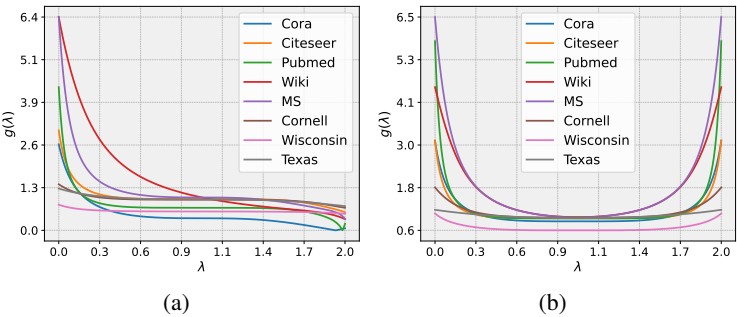

(a)                                                          (b)

Figure 4: The graph filters learned on different data sets, with the parameter $P$ being odd in (a) and even in (b).

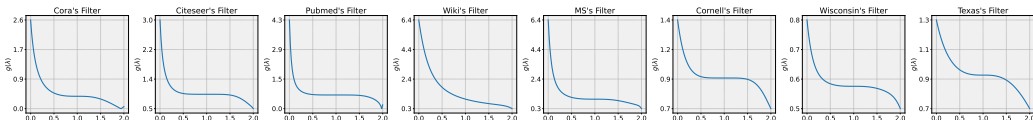

Figure 5: The graph filters learned using different data sets, with parameter $P$ being odd.

## A.3  P-HOP FILTER

Figure 7 shows the performance associated with different $P$ on Cora, Citeseer, and Pubmed datasets.

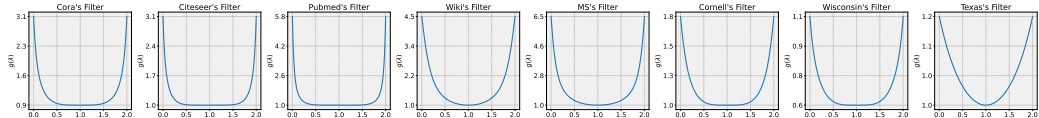

Figure 6: The graph filters learned using different data sets, with parameter $P$ being even.

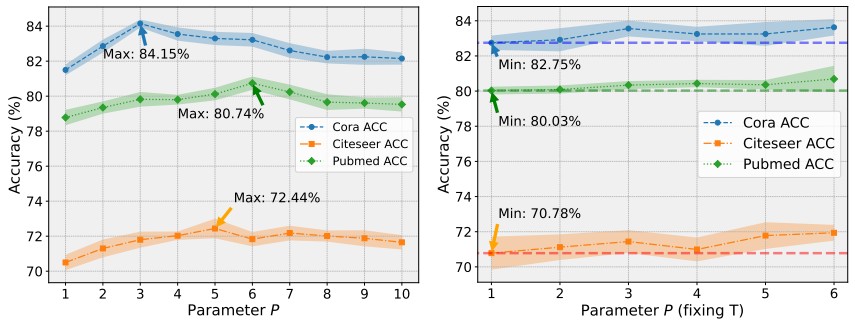

Figure 7: Performance impact on Cora, Citeseer and Pubmed of increasing $P$ in two scenarios (a) when $K$ is fixed, (b)when $T = KP$ is fixed.

### A.4 TREND OF LEARNABLE COEFFICIENTS

As shown in Figure 8, we can observe that the learned weight $\beta$ tends to have the same sign on homophilic datasets, but not on heterophilic datasets. That is because the graph filter can adaptively learn an opposite coefficient for improper propagation. The tendency of the coefficient can further gain insight into the homophily and heterophily of the graph.

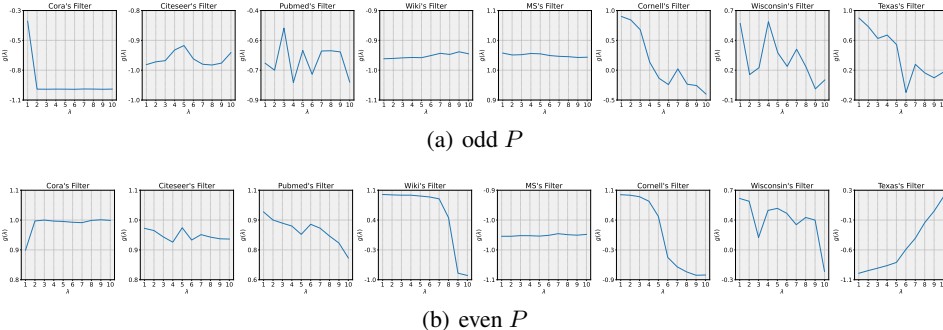

Figure 8: The learned weights on different datasets when $P$ is (a) odd or (b) even.

### A.5 HYPERPARAMETERS SETTINGS

The hyperparameters settings for different parity of $P$ are shown in Table 3 and Table 4.

### A.6 PROOF FOR LEMMA 1

($\Rightarrow$). We show the result by contradiction. If $\sum_k^\infty |a_k|$ is not convergent, then at $\gamma = 1$, we have $\sum_k^\infty |a_k \gamma^k|$ is not convergent, which occurs a contradiction. Therefore, series $\sum_k^\infty |a_k|$ converges.

($\Leftarrow$). It is obvious that for $\forall \gamma \in (-1, 1]$, $\sum_k^\infty |a_k \lambda^k| \leq \sum_k^\infty |a_k|$. Therefore, $\sum_k^\infty |a_k \lambda^k|$ uniformly converges in $\lambda \in (-1, 1]$. $\qquad \square$

Table 3: The hyperparameters of APGNN on various datasets when parameter $P$ is odd.

| Dataset | $K$ | $P$ | $\alpha$ | Weight decay | Learning rate | Dropout rate |
|---------|-----|-----|----------|--------------|---------------|--------------|
| **Cora** | 10 | 3 | 0.7 | 0.005 | 0.01 | 0.8 |
| **Citeseer** | 10 | 5 | 0.7 | 0.00625 | 0.01 | 0.5 |
| **Pubmed** | 10 | 5 | 0.9 | 0.005 | 0.01 | 0.5 |
| **Wiki-CS** | 10 | 1 | 0.9 | 0.000525 | 0.03 | 0.4 |
| **MS-Academic** | 10 | 3 | 0.9 | 0.00525 | 0.02 | 0.4 |
| **Cornell** | 10 | 3 | 0.6 | 0.001 | 0.01 | 0.2 |
| **Wisconsin** | 10 | 3 | 0.2 | 0.001 | 0.01 | 0.2 |
| **Texas** | 10 | 3 | 0.1 | 0.001 | 0.01 | 0.2 |

Table 4: The hyperparameters of APGNN on various datasets when parameter $P$ is even.

| Dataset | $K$ | $P$ | $\alpha$ | Weight decay | Learning rate | Dropout rate |
|---------|-----|-----|----------|--------------|---------------|--------------|
| **Cora** | 10 | 4 | 0.7 | 0.005 | 0.01 | 0.8 |
| **Citeseer** | 10 | 6 | 0.7 | 0.00625 | 0.01 | 0.5 |
| **Pubmed** | 10 | 6 | 0.9 | 0.005 | 0.01 | 0.5 |
| **Wiki-CS** | 10 | 2 | 0.9 | 0.000525 | 0.03 | 0.4 |
| **MS-Academic** | 10 | 2 | 0.9 | 0.00525 | 0.02 | 0.4 |
| **Cornell** | 10 | 2 | 0.6 | 0.001 | 0.01 | 0.2 |
| **Wisconsin** | 10 | 2 | 0.2 | 0.001 | 0.01 | 0.2 |
| **Texas** | 10 | 2 | 0.1 | 0.001 | 0.01 | 0.2 |

### A.7 PROOF FOR THEOREM 1

$\tilde{\mathbf{A}}$ is an adjacency matrix of a graph, which is a real symmetric matrix. Since we can decompose $\tilde{\mathbf{A}}$ as $\tilde{\mathbf{A}} = \mathbf{U}\boldsymbol{\Gamma}\mathbf{U}^\top$, where $\mathbf{U}$ is a matrix composed of the eigenvectors of $\tilde{\mathbf{A}}$ and $\boldsymbol{\Gamma} = \mathrm{diag}(\gamma_1, \cdots, \gamma_n)$ is the diagonal matrix of the corresponding eigenvalues. Therefore, we have

$$g(\mathbf{L}) = \sum_{k=1}^{\infty} \theta_k \tilde{\mathbf{A}}^k = \mathbf{U}\mathrm{diag}\left(\sum_{k=1}^{\infty} \theta_k \gamma_1^k, \ \cdots, \ \sum_{k=1}^{\infty} \theta_k \gamma_n^k\right)\mathbf{U}^\top \tag{24}$$

Therefore, the $g(\mathbf{L})$ converges absolutely and uniformly if and only if $\sum_{k=1}^{\infty} \theta_k \gamma_i^k$ converges absolutely and uniformly for all $i \in [n]$. Then apply Lemma 1 and we can obtain the result. $\qquad\square$

### A.8 PROOF OF THEOREM 2

We first introduce some definitions and Lemma for assisting with the proof.

**Definition 1.** Consider the sample set $S = \{\mathbf{x}_1, \cdots, \mathbf{x}_n\}$ and function set $\mathcal{F}$, where $f(\mathbf{x})$ is bounded for any $f \in \mathcal{F}$. Then the empirical Rademacher complexity is defined as:

$$\Re_S(\mathcal{F}) = \frac{1}{n}\mathbb{E}_{\boldsymbol{\sigma}}\left[\sup_{f \in \mathcal{F}} \sum_{i=1}^{n} \sigma_i f(\mathbf{x}_i)\right], \tag{25}$$

where $\sigma_i$ is i.i.d. Rademacher random variable defined by $\Pr(\sigma_i = -1) = \Pr(\sigma_i = 1) = 0.5$.

**Lemma 2.** *Consider the hypothesis set*

$$\mathcal{H}_\mathcal{X} = \{h : h(\mathbf{x}) = g_{\boldsymbol{\theta}} Lf(\mathbf{x}), \ f(\mathbf{x}) = \langle \mathbf{w}, \mathbf{x} \rangle, \ \|\mathbf{w}\|_2 \le B, \ \|\boldsymbol{\theta}\|_1 \le M\}, \tag{26}$$

*where $\boldsymbol{\theta} = [\theta_0, \theta_1, \cdots, \theta_K]$. Let $x_j$ denote the $j$-th element of $\mathbf{x} \in \mathcal{X}$, and $\mathbb{E}[x_j^2] \le c_\mathcal{X}$ for any $j \in [d]$. Then for any sample set $S = \{\mathbf{x}_1, \cdots, \mathbf{x}_{n_l}\} \subset \mathcal{X}$ we have*

$$\Re_S(\mathcal{F}_\mathcal{X}) \lesssim 2BMc_\mathcal{X}\sqrt{\frac{2\log(2K+2)}{n_l}}. \tag{27}$$

**Lemma 3** (Massart's Lemma (Mohri et al., 2018)). *Let $\mathcal{X} \subset \mathbb{R}^n$ be a finite set and $\sup_{\mathbf{x} \in \mathcal{X}} \|\mathbf{x}\|_2 \le r\sqrt{n}$, then the following inequality holds:*

$$\mathbb{E}_{\boldsymbol{\sigma}}\left[\frac{1}{n}\langle\boldsymbol{\sigma}, \mathbf{x}\rangle\right] \le r\sqrt{\frac{2\log|\mathcal{X}|}{n}}, \tag{28}$$

*where $\boldsymbol{\sigma} = [\sigma_1, \cdots, \sigma_n]$ denote the vector of Rademacher random variables.*

*Proof.* Based on the definition, we can write

$$
\begin{aligned}
\mathfrak{R}_S(\mathcal{H}_\mathcal{X}) &= \frac{1}{n_l}\mathbb{E}_{\boldsymbol{\sigma}}\left[\sup_{h_{\mathbf{w},\boldsymbol{\theta}}\in\mathcal{H}_\mathcal{X}}\sum_{i=1}^{n_l}\sigma_i h_{\mathbf{w},\boldsymbol{\theta}}(\mathbf{x}_i)\right]\\
&= \frac{1}{n_l}\mathbb{E}_{\boldsymbol{\sigma}}\left[\sup_{\|\mathbf{w}\|_2\le B,\ \|\boldsymbol{\theta}\|_1\le M}\sum_{i=1}^{n_l}\sigma_i g_{\boldsymbol{\theta}}Lf(\mathbf{x}_i)\right]\\
&= \frac{1}{n_l}\mathbb{E}_{\boldsymbol{\sigma}}\left[\sup_{\|\mathbf{w}\|_2\le B,\ \|\boldsymbol{\theta}\|_1\le M}\sum_{i=1}^{n_l}\sigma_i\int_\mathcal{X}\sum_{k=0}^K\theta_k\tilde{A}^k(\mathbf{x}_i,\mathbf{x})\mathbf{x}^\top\mathbf{w}\mathrm{d}p(\mathbf{x})\right]\\
&\le \frac{B}{n_l}\mathbb{E}_{\boldsymbol{\sigma}}\left[\sup_{\|\boldsymbol{\theta}\|_1\le M}\left\|\sum_{i=1}^{n_l}\sigma_i\int_\mathcal{X}\sum_{k=0}^K\theta_k\tilde{A}^k(\mathbf{x}_i,\mathbf{x})\mathbf{x}\mathrm{d}p(\mathbf{x})\right\|_2\right]\\
&= \frac{B}{n_l}\mathbb{E}_{\boldsymbol{\sigma}}\left[\sup_{\{\mathbf{v}_i\}_{i=1}^n\in V}\left\|\sum_{i=1}^{n_l}\sigma_i\mathbf{v}_i\right\|_2\right]
\end{aligned}
$$

where the inequality follows from the Cauchy-Schwarz inequality, and the set $V$ is defined as

$$V \triangleq \left\{\{\mathbf{v}_i\}_{i=1}^n : \mathbf{v}_i = \int_\mathcal{X}\sum_{k=0}^K\theta_k\tilde{A}^k(\mathbf{x}_i,\mathbf{x})\mathbf{x}\mathrm{d}p(\mathbf{x}),\quad \|\boldsymbol{\theta}\|_1\le M\right\}. \tag{29}$$

Define $q_j(\mathbf{x}) = x_j$ returning the $j$-th coordinate of the input. Hence, the $j$-th coordinate of $\mathbf{v}_i$ can be rewritten as

$$v_{ij} = \int_\mathcal{X}\sum_{k=0}^K\theta_k\tilde{A}^k(\mathbf{x}_i,\mathbf{x})x_j\mathrm{d}p(\mathbf{x}) = \int_\mathcal{X}\sum_{k=0}^K\theta_k\tilde{A}^k(\mathbf{x}_i,\mathbf{x})q_j(\mathbf{x})\mathrm{d}p(\mathbf{x}) = \sum_{k=0}^K\theta_k\tilde{A}^k q_j(\mathbf{x}_i). \tag{30}$$

Since $\|\mathbf{u}\|_2 \le \sqrt{d}\|\mathbf{u}\|_\infty$ for any $\mathbf{u} \in \mathbb{R}^d$, we have

$$
\begin{aligned}
\mathfrak{R}_S(\mathcal{H}_\mathcal{X}) &\le \frac{B\sqrt{d}}{n_l}\mathbb{E}_{\boldsymbol{\sigma}}\left[\sup_{\{\mathbf{v}_i\}_{i=1}^n\in V}\left\|\sum_{i=1}^{n_l}\sigma_i\mathbf{v}_i\right\|_\infty\right]\\
&\le \frac{B\sqrt{d}}{n_l}\mathbb{E}_{\boldsymbol{\sigma}}\left[\sup_{\{\mathbf{v}_i\}_{i=1}^n\in V}\max_{j\in[d]}\left|\sum_{i=1}^{n_l}\sigma_i v_{ij}\right|\right]\\
&\le \frac{2B\sqrt{d}}{n_l}\mathbb{E}_{\boldsymbol{\sigma}}\left[\sup_{\{\mathbf{v}_i\}_{i=1}^n\in V}\max_{j\in[d]}\sum_{i=1}^{n_l}\sigma_i v_{ij}\right]\\
&\le \frac{2B\sqrt{d}}{n_l}\mathbb{E}_{\boldsymbol{\sigma}}\left[\sup_{\|\boldsymbol{\theta}\|_1\le M}\sum_{i=1}^{n_l}\sigma_i\sum_{k=0}^K\theta_k\tilde{A}^k q_j(\mathbf{x}_i)\right]\\
&= \frac{2B\sqrt{d}}{n_l}\mathbb{E}_{\boldsymbol{\sigma}}\left[\sup_{\|\boldsymbol{\theta}\|_1\le M}\sum_{k=0}^K\theta_k\sum_{i=1}^{n_l}\sigma_i\tilde{A}^k q_j(\mathbf{x}_i)\right]\\
&= \frac{2BM\sqrt{d}}{n_l}\mathbb{E}_{\boldsymbol{\sigma}}\left[\sup_{\boldsymbol{\theta}\in\Theta}\sum_{k=0}^K\theta_k\sum_{i=1}^{n_l}\sigma_i\tilde{A}^k q_j(\mathbf{x}_i)\right]\\
&= 2BM\sqrt{d}\mathfrak{R}_S(\mathcal{H}'),
\end{aligned}
$$

where $\boldsymbol{\Theta} = \bigcup_{k=0}^{K}\{-\boldsymbol{e}_k, \boldsymbol{e}_k\}$ and $\boldsymbol{e}_k$ denote $k$-th vector with $k$-th entry as one and others are zero. The set $\mathcal{H}' = \{h(\mathbf{x}) = \sum_{k=0}^{K} \theta_k A^k q_j(\mathbf{x}) : \boldsymbol{\theta} \in \boldsymbol{\Theta}\}$ is a finite set with $|\mathcal{H}'| = 2(K+1)$. We bound $\mathfrak{R}_S(\mathcal{H}')$ with Lemma 3:

Since $\mathcal{H}'$ is a finite set and for any $h \in \mathcal{H}'$,

$$\frac{1}{n_l}\sum_{i=1}^{n_l} h(\mathbf{x}_i)^2 = \frac{1}{n_l}\sum_{i=1}^{n_l} \sup_{k\in[n]}\left[\tilde{A}^k q_j(\mathbf{x}_i)\right]^2$$

$$\approx \int_{\mathcal{X}} \sup_{k\in[n]}\left[\tilde{A}^k q_j(\mathbf{x})\right]^2 \mathrm{d}p(\mathbf{x}) \leq \|q_j\|^2 \leq c_{\mathcal{X}}^2.$$

where we use $\|\tilde{A}^k q_j\| \leq \|\tilde{A}^k\|\|q_j\|$ and $\|\tilde{A}^k\| \leq 1$ for any $k \in [n]$, and

$$\|q_j\|^2 = \int_{\mathcal{X}} q_j(\mathbf{x})^2 \mathrm{d}p(\mathbf{x}) = \int_{\mathcal{X}} x_j^2 \mathrm{d}p(\mathbf{x}) = \mathbb{E}[x_j^2] \leq c_{\mathcal{X}}^2. \tag{31}$$

Therefore we finally obtain

$$\mathfrak{R}_S(\mathcal{F}_{\mathcal{X}}) \lesssim 2BMc_{\mathcal{X}}\sqrt{\frac{2d\log(2K+2)}{n_l}} \tag{32}$$

As a remark, we can present a more precise bound through McDiarmid's inequality. consider the convergence

$$\frac{1}{n_l}\sum_{i=1}^{n_l} \sup_{k\in[n]}\left[\tilde{A}^k q_j(\mathbf{x}_i)\right]^2 \to \int_{\mathcal{X}} \sup_{k\in[n]}\left[\tilde{A}^k q_j(\mathbf{x})\right]^2 \mathrm{d}p(\mathbf{x}). \tag{33}$$

With the probability of at least $1 - \delta$,

$$\frac{1}{n_l}\sum_{i=1}^{n_l} \sup_{k\in[n]}\left[\tilde{A}^k q_j(\mathbf{x}_i)\right]^2 \leq \int_{\mathcal{X}} \sup_{k\in[n]}\left[\tilde{A}^k q_j(\mathbf{x})\right]^2 \mathrm{d}p(\mathbf{x}) - \mathcal{O}\left(\sqrt{\frac{\log 1/\delta}{n_l}}\right). \tag{34}$$

The details are omitted since it is not the major part of the analysis. $\qquad\square$

*Proof.* We first write the excess risk decomposition:

$$R(h_{\mathbf{w},\boldsymbol{\theta}}) - \hat{R}(\hat{h}_{\mathbf{w},\boldsymbol{\theta}}) = \underbrace{R(h_{\mathbf{w},\boldsymbol{\theta}}) - \hat{R}(h_{\mathbf{w},\boldsymbol{\theta}})}_{A\text{ part}} + \underbrace{\hat{R}(h_{\mathbf{w},\boldsymbol{\theta}}) - \hat{R}(\hat{h}_{\mathbf{w},\boldsymbol{\theta}})}_{B\text{ part}} \tag{35}$$

For the $A$ part, we first apply Theorem 5.8 in (Mohri et al., 2018). With probability at least $1 - \delta$,

$$R(h_{\mathbf{w},\boldsymbol{\theta}}) - \hat{R}(h_{\mathbf{w},\boldsymbol{\theta}}) \leq \mathfrak{R}_S(\mathcal{H}_{\mathcal{X}}) + 3\sqrt{\frac{\log(2/\delta)}{2n_l}}. \tag{36}$$

Since the last term is of order $\mathcal{O}(\sqrt{\log(1/\delta)n_l^{-1}})$, which is significantly smaller than $\mathfrak{R}_S(\mathcal{H}_{\mathcal{X}})$, we rewrite the above inequality as

$$R(h_{\mathbf{w},\boldsymbol{\theta}}) \lesssim \hat{R}(h_{\mathbf{w},\boldsymbol{\theta}}) + 2BMc_{\mathcal{X}}\sqrt{\frac{2d\log(2K+2)}{n_l}}, \tag{37}$$

where we replace the Rademacher complexity with its upper bound by Lemma 2.

For the $B$ part, we first define the empirical operator over $S = \{\mathbf{x}_1, \cdots, \mathbf{x}_n\}$,

$$L_n f = \frac{1}{n}\sum_{i=1}^{n} \frac{A(\mathbf{x}_i, \cdot)}{\sqrt{d_n(\mathbf{x}_i)d_n(\cdot)}} f(\mathbf{x}_i), \quad d_n = \frac{1}{n}\sum_{i=1}^{n} A(\mathbf{x}_i, \cdot) \tag{38}$$

and $g_{\boldsymbol{\theta}} L_n = \sum_{k=0}^{K} \theta_k L_n^k$. Then we have

$$\hat{R}(h_{\mathbf{w},\boldsymbol{\theta}}) - \hat{R}(\hat{h}_{\mathbf{w},\boldsymbol{\theta}}) \leq \left|\hat{R}(h_{\mathbf{w},\boldsymbol{\theta}}) - \hat{R}(\hat{h}_{\mathbf{w},\boldsymbol{\theta}})\right|$$

$$\leq \frac{1}{n_l} \left| \sum_{i=1}^{n_l} h_{\mathbf{w},\boldsymbol{\theta}}(\mathbf{x}_i) - \hat{h}_{\mathbf{w},\boldsymbol{\theta}}(\mathbf{x}_i) \right|$$

$$= \frac{1}{n_l} \left| \sum_{i=1}^{n_l} g_{\boldsymbol{\theta}} L f(\mathbf{x}_i) - g_{\boldsymbol{\theta}} L_n f(\mathbf{x}_i) \right|$$

$$\leq \frac{1}{n_l} \left[ \sum_{i=1}^{n_l} \left( g_{\boldsymbol{\theta}} L f(\mathbf{x}_i) - g_{\boldsymbol{\theta}} L_n f(\mathbf{x}_i) \right)^2 \right]^{1/2}$$

$$\approx \| g_{\boldsymbol{\theta}} L f - g_{\boldsymbol{\theta}} L_n f \|$$

$$\leq \| g_{\boldsymbol{\theta}} L - g_{\boldsymbol{\theta}} L_n \| \| f \|.$$

where the second inequality follows from the Lipschitz property. With Cauchy-Schwarz inequality,

$$\| f \|_2 = \int_{\mathcal{X}} \langle \mathbf{w}, \mathbf{x} \rangle \mathrm{d}p(\mathbf{x}) \leq B \cdot \mathbb{E}_{\mathbf{x}}[\|\mathbf{x}\|_2] \leq B d c_{\mathcal{X}}. \tag{39}$$

According to Theorem 15 of (Rosasco et al., 2010), there exists a proper constant $C > 0$ related to $A(\cdot, \cdot)$, such that

$$\| L - L_n \| \leq \| L - L_n \|_{HS} \leq C \sqrt{\frac{\log(2/\delta)}{n}}. \tag{40}$$

with probability at least $1 - \delta$. Since the polynomial $g_{\boldsymbol{\theta}}$ is $L_M$-Lipschitz, we have

$$\| g_{\boldsymbol{\theta}} L - g_{\boldsymbol{\theta}} L_n \| \leq L_M C \sqrt{\frac{\log(2/\delta)}{n}} \tag{41}$$

Combining the above results, one can conclude that for any $(h_{\mathbf{w},\boldsymbol{\theta}}, \hat{h}_{\mathbf{w},\boldsymbol{\theta}}) \in \mathcal{H}_{\mathcal{X}} \times \mathcal{H}_S$,

$$R(h_{\mathbf{w},\boldsymbol{\theta}}) \lesssim \hat{R}(\hat{h}_{\mathbf{w},\boldsymbol{\theta}}) + 2BMc_{\mathcal{X}} \sqrt{\frac{2d \log(2K+2)}{n_l}} + BCL_M d c_{\mathcal{X}} \sqrt{\frac{\log(2/\delta)}{n}}. \tag{42}$$

with probability at least $1 - \delta$. $\qquad\square$

### A.9 COMPUTATIONAL COMPLEXITY

Denote $N$ as the number of nodes, $E$ as the set of edges, $d$ as the hidden layer of MLP, and $c$ as the number of classes. The adjacency matrix is stored in sparse format (i.e., only the edge will be stored), so the space complexity is $O(|E|)$. In our implementation, we use a 2-layer MLP for feature extraction and $K$-order polynomial graph filter. Therefore, the time complexity of MLP and graph convolution is $O(NdcL)$ and $O(Kc|E|)$, respectively. Therefore, the complexity grows linearly as the number of samples increases, and it also depends on the number of edges in the graph.

