# OpenReview forum: "Towards the Universal Learning Principle for Graph Neural Networks"
_ICLR.cc/2024/Conference — Submitted to ICLR 2024_

### Official Review · Reviewer_W4db · 2023-10-31

**Soundness:** 2 fair
**Presentation:** 1 poor
**Contribution:** 1 poor
**Rating:** 3
**Confidence:** 3

**Summary:**

This paper reformulates the parameters of DAGNN from $\sum_{k=0}^K\theta_k\tilde A^k$ to $\sum_{k=0}^K\beta_k\alpha^k\tilde A^{kP}$, where $|\beta_k|<1, \alpha\in(0,1)$, and $P\in\mathbb N$ is a hyper-parameter for skipping hops.  They show improved performance in their specific evaluation procedure on several datasets.

There are generalization error bounds with bounded $K$ and the continuous and L-Lipschitz smoothness assumptions. They also show that this parameterization format won't explode to infinity as $K\rightarrow \inf$ due to $\alpha^k$. The function is also Lipschitz's continuous.

**Strengths:**

The P-hop filter adds one more hyper-parameter to tune for, e.g., DAGNN. This additional degree of freedom shows improved performance in the specific evaluation procedure on these datasets.

The authors dedicated efforts to deriving a generalization error bound of GNNs under the specific assumptions.

**Weaknesses:**

The method is not as novel, and many of the analyses are unrelated and unnecessary.
* The infinite-depth discussion and analysis seem to be unnecessary and could be overclaiming. The actually used APGNN filter still has truncated fixed limited depth. As stated in the paper, "In practice, we generally set K < n since the neighbor information beyond n-hops is redundant, restricting the complexity away from infinity.". After all, the graph size is limited and finite anyway.
* Similarly, the "stability through Lipstichz smoothness" "principle" for designing all GNNs is too strong and may not be necessary in general. The choice of GNNs depends more on the specific data distribution to approximate.
* The resulting APGNN model is similar to the previous DAGNN except for the concrete parameterizations of $\theta^k$.

The evaluation procedure is also concerned with potentially unfair comparison.
* There seems to be no cross-validation procedure to tune hyper-parameters even though many of the datasets are small (at most 1000 test nodes).
* The efforts for tuning hyper-parameters seem non-equal between proposed methods and baselines.

**Questions:**

* Are there results with more carefully designed evaluation procedures? For example, use 10-fold cross-validation to tune hyper-parameters, and then only apply the selected hyper-parameter once to report the performances. Assigning equal hyper-parameter tuning efforts to all baselines and proposed methods is desirable for a fairer comparison. It is okay for the performances to be lower than other reported ones, as long as they are reliable, reproducible, and comparable. It would also be better to include other larger datasets for evaluation even though the proposed methods' performances may not be better than baselines on them, as long as there are reasonable explanations.

---

### Official Review · Reviewer_2jyF · 2023-10-31

**Soundness:** 3 good
**Presentation:** 3 good
**Contribution:** 2 fair
**Rating:** 3
**Confidence:** 3

**Summary:**

This paper derives theoretical conditions on the coefficients of polynomial graph filters formed by power series to ensure convergence and Lipschitz continuity as the depth goes to infinity. This provides guidance on constructing deeper GNNs. This paper also proposes Adaptive Power GNN (APGNN) following the proposed principle. APGNN uses exponentially decaying weights to aggregate information of different orders and a P-hop filter to perceive deeper neighbor information. This allows seamless extension to infinite depths. Finally, this paper empirically shows that APGNN achieves state-of-the-art performance on node classification tasks across both homophilic and heterophilic graphs.

**Strengths:**

(1) This paper is generally well-organized and easy to follow.

(2) This paper provides abundant theoretical analysis, which is helpful in soliciting the design of APGNN.

(3) The superiority exhibited in the experimental results seems promising. However, more baselines are supposed to be involved. See more details in weaknesses and questions.

**Weaknesses:**

(1) The motivation of this paper is unconvincing. This paper claims that the motivation lies in, first, "the convergence of graph filters can not be guaranteed"; second, we need "the construction principle of an infinite deep GNN". However, for the first point, the authors did not give any supporting evidence on the non-convergence; for the second point, there is already a series of existing works on over-smoothing revealing how to build deep GNNs, and I personally believe that they have already revealed abundant construction principles for deep GNNs. In addition, the author should also clarify why we need to build infinite deep GNNs instead of regular deep GNNs.

(2) The author should present the connection and differences between this work and the series works of deep GNNs.

(3) Since there is already a series of excellent deep GNNs, more baselines should be adopted, e.g., GCNII [1].

[1] Chen, M., Wei, Z., Huang, Z., Ding, B., & Li, Y. (2020, November). Simple and deep graph convolutional networks. In International conference on machine learning (pp. 1725-1735). PMLR.

**Questions:**

(1) By "graph filters", do the authors mean the whole model including the learnable parameters of GNNs? Why does the convergence of graph filters (may) become problematic when the layer number goes larger?

(2) There has been a series of existing works on over-smoothing revealing how to build deep GNNs, and I personally believe that they have already revealed abundant construction principles for deep GNNs. What else principle do we need to know about regarding deep GNNs?

(3) Why do we need infinitely deep GNNs? Many deep GNNs have already achieved exceptional performances. It would be much better if more deep GNNs, such as GCNII, could be adopted as baselines.

---

### Official Review · Reviewer_X3Vf · 2023-10-31

**Soundness:** 3 good
**Presentation:** 3 good
**Contribution:** 2 fair
**Rating:** 3
**Confidence:** 4

**Summary:**

This paper introduces a polynomial graph filter with specialized coefficients. It provides theoretical claims of convergence and stability. The overall explanations are lucid, and the experiments are presented to validate the improvements.

**Strengths:**

This paper delivers a thorough analysis of both existing and newly proposed graph filters. It presents theoretical proof to support its claims. Additionally, the paper extends its analysis to include the generalization from discrete graphs to continuous graphs, adding to the comprehensiveness of the research. The experiments are presented to show some improvements.

**Weaknesses:**

1. The novelty of the proposed polynomial graph filter is a bit limited. The incorporation of the α^k term is similar to the decay rate used in PPNP, and the inclusion of a learnable parameter β (although not explicitly described in the paper) can draw strong similarities with GPR-GNN and many other polynomial filter methods such as ChebNet, ChebNetII, and BerNet. The primary novelty stems from the design of the P-hop filter, which essentially uses the power of the adjacent matrix as the new adjacent matrix to capture longer distance relations with significantly higher computation and memory costs. The paper refers to its proposed method as the "adaptive power GNN", but many existing works have already achieved the same goal.

2. The motivation behind the proposed method is somewhat unclear. The design of the exponential decreasing coefficients lacks a clearer explanation and motivation. While some strength such as convergence and stability is claimed, there is no clear evidence why the proposed one can be better than existing ones such as GPR-GNN which also has convergence and stability based on the description in Section 4.2. Therefore, it would be beneficial to provide a more in-depth explanation of the intuition behind the design to enhance understanding.

3. The experiment does not provide sufficient ablation study to investigate why the proposed method can improve the performance over existing polynomial methods. Additional explanations and in-depth analysis of the experiments are anticipated.

4. The experimental comparisons with the baselines are unfair. Specifically, it is mentioned in section 6.1 that it fixes the polynomial order K=10 for baselines such as ChebNet, APPNP, GNN-LF, GNN-HF, DAGNN, APPNP, GPR-GNN, and BernNet. However, the proposed method selects K in the range {1,2,..., 20} and the total number of propagation layers is T=KP where P is selected from the range {1,2,3,4,5}. In other words, the proposed method uses significantly more layers than the baselines. This raises concerns about the improvement over baselines.

5. The paper claims that different from existing GNNs, APGNN can be seamlessly extended to an infinite-depth network. However, the method still chooses a finite-depth implementation, and there is no evidence on how to extend it to an infinite-depth network.

**Questions:**

Please refer to the weakness.

---

### Meta-Review · Area_Chair_vQ74 · 2023-12-06

**Metareview:**

The paper studies the convergence and stability of graph filters in GNNs under infinite-depth scenarios, and proposes a criterion for the graph filter formed by power series. However, all reviewers have shown serious concerns on the below points:

- The position of this paper is not clear. Its motivation is not convincing and its in-depth analysis can be overclaiming. The differences between this work and the series works of deep GNNs should be discussed.
- The evaluation procedure is also concerned with potentially unfair comparisons, and more baselines should be adopted, e.g., GCNII [1].

**Justification For Why Not Higher Score:**

Clearly limited novelty and problems in comparisons with baselines.

**Justification For Why Not Lower Score:**

N/A.

---

### Decision · Program_Chairs · 2024-01-16

Reject